# Stigmatization from Work-Related COVID-19 Exposure: A Systematic Review with Meta-Analysis

**DOI:** 10.3390/ijerph18126183

**Published:** 2021-06-08

**Authors:** Melanie Schubert, Julia Ludwig, Alice Freiberg, Taurai Monalisa Hahne, Karla Romero Starke, Maria Girbig, Gudrun Faller, Christian Apfelbacher, Olaf von dem Knesebeck, Andreas Seidler

**Affiliations:** 1Institute and Policlinic of Occupational and Social Medicine, Faculty of Medicine, Technische Universität Dresden, 01307 Dresden, Germany; alice.freiberg@tu-dresden.de (A.F.); karla.romero_starke@tu-dresden.de (K.R.S.); maria.girbig@tu-dresden.de (M.G.); andreas.seidler@tu-dresden.de (A.S.); 2Institute of Medical Sociology, University Medical Center Hamburg-Eppendorf, 20246 Hamburg, Germany; j.ludwig@uke.de (J.L.); o.knesebeck@uke.de (O.v.d.K.); 3Institute of Social Medicine and Health Systems Research, Otto von Guericke University Magdeburg, 39120 Magdeburg, Germany; taurai.hahne@med.ovgu.de (T.M.H.); christian.apfelbacher@med.ovgu.de (C.A.); 4Institute of Sociology, Faculty of Behavioural and Social Sciences, Chemnitz University of Technology, Thüringer Weg 9, 09126 Chemnitz, Germany; 5Department of Community Health, Hochschule für Gesundheit, 44801 Bochum, Germany; gudrun.faller@hs-gesundheit.de

**Keywords:** corona, COVID-19, bullying, discrimination, healthcare workers, nursing, SARS-CoV2, stigma, work

## Abstract

Stigmatization from work-related COVID-19 exposure has not been investigated in detail yet. Therefore, we systematically searched three databases: Medline, Embase, and PsychInfo (until October 2020), and performed a grey literature search (until February 2021). We identified 46 suitable articles from 24 quantitative and 11 qualitative studies, 6 systematic reviews, 3 study protocols and 1 intervention. The assessment of stigmatization varied widely, ranging from a single-item question to a 22-item questionnaire. Studies mostly considered perceived self-stigma (27 of 35 original studies) in healthcare workers (HCWs) or hospital-related jobs (29 of 35). All articles reported on stigmatization as a result of work-related COVID-19 exposure. However, most quantitative studies were characterized by convenience sampling (17 of 24), and all studies—also those with an adequate sampling design—were considered of low methodological quality. Therefore, it is not possible to determine prevalence of stigmatization in defined occupational groups. Nevertheless, the work-related stigmatization of occupational groups with or without suspected contact to COVID-19 is a relevant problem and increases the risk for depression (odds ratio (OR) = 1.74; 95% confidence interval CI 1.29–2.36) and anxiety (OR = 1.75; 95% CI 1.29–2.37). For promoting workers’ health, anti-stigma strategies and support should be implemented in the workplace.

## 1. Introduction

The COVID-19 pandemic has had a dramatic impact on all aspects of our lives. Besides the direct effects of the pandemic, there are also many indirect social consequences. People’s everyday life is disrupted and negatively affected by the pandemic. Compliance with the hygiene measures and in particular the “infodemic”, which is characterized by an overabundance of news covering facts, rumours and misinformation [1], have triggered or strengthened negative feelings such as fear, anger and hatred in the population. International and national media report on a “witch-hunt hysteria”, attacks on index cases, infected people and relatives, and stigmatization [2,3].

The stigmatization process includes labelling a person with a specific characteristic, linking the label to undesirable characteristics (stereotypes) which result in negative emotional reactions. This constitutes the basis for the separation of “us” and “them”, and leads to discrimination and the loss of status [4]. Furthermore, the stigmatization process depends on the social, political and economic power of the stigmatized group [4].

On the action-oriented level of stigmatization, a distinction between public stigma, stigma by association and self-stigma can be made [5]. Thereby, endorsed stereotypes, prejudice, and discrimination in the general public against a particular group are defined as public stigma [6]. When this stigmatization process is transferred to relatives of the stigmatized individuals, it is referred to as associative stigma [7]. Self-stigmatization, on the other hand, is defined as the internalization of the very same stereotypes and prejudices [6] and can be additionally divided into perceived (belief that “most” people devalue and discriminate against the individuals holding the specific characteristic) [8] and anticipated stigma (expectation of experiencing prejudice and discrimination based on the characteristic) [9].

The consequences of stigmatization can be diverse for those affected and include discrimination-related stress, reduced self-esteem and reduced self-efficacy [10]. Stigmatization can lead to a reduced quality of life [11], mental illnesses, trauma, and even suicide [12,13,14,15]. Furthermore, the stigma may not be disclosed, which is associated with less use of professional help, and poorer assessment of one’s own health [16]. As a result, the feared stigmatization by the public may lead to a concealment of the disease [9,17], and may increase the spread of it. In association with work, stigmatized employees are less able to cope with daily work demands [12], and also report lower job satisfaction, job performance, work commitment, and willingness to learn and develop [18,19]. Furthermore, stigmatization can also initiate or solidify career-related consequences such as decline [20], dismissals or “voluntary” dismissals [21].

Workers exposed to potential hazards and diseases such as COVID-19 may also suffer from stigmatization. Stigmatization is a key feature of bullying [22,23]. Leymann [22] and Einarsen [24] identified four phases: aggressive behaviour, bullying, stigma and severe trauma. The initial phase is characterized by indirect and direct aggressive behaviour which leaves the affected person humiliated and increasingly isolated. The persons become more and more stigmatized, making it more difficult to for them to defend themselves. Stigmatization makes it less possible for the person concerned to cope with daily work demands [12]. There are also associations with lower job satisfaction, job performance, work commitment, willingness to learn [18,19]. Stigmatization can also initiate or consolidate the relegation of careers [20] and dismissals or “voluntary” dismissals [21].

Stigmatization can result in an additional high psychological stress for workers. Recent systematic reviews indicate that stigmatization is a risk factor for mental disorders in healthcare workers (HCWs) caring for MERS-CoV2/SARS-patients [25,26]. Recently, a large number of systematic reviews investigating the psychological impact of work-related COVID-19 exposure such as stigmatization have been published. However, these reviews only include no or very few original studies related to COVID-19 [27,28,29,30,31,32,33,34]. Thus far, a systematic review of the literature on work-related stigmatization focussing on COVID-19 is missing. The aim of this systematic review was to provide a comprehensive overview of COVID-19-related stigmatization across occupational classes. Furthermore, we aimed to summarize health consequences of work-related stigmatization from COVID-19 exposure using a meta-analytic approach.

## 2. Materials and Methods

### 2.1. Review Questions and Study Eligibility Criteria

We systematically searched for publications on the stigmatization experience of employees due to COVID-19. Our main question was whether employees experience stigmatization in association with work due to COVID-19. Related to the main objective, we were interested in describing the specific forms of stigmatization (e.g., public stigma, self-stigma, associative stigma), how often they were observed in specific occupational groups, and if employees returning to work (such as after quarantine, or infection) experience stigmatization (A). Furthermore, we aimed to identify health consequences as a result of work-related stigmatization due to COVID-19 (B), and organizational or other conditions that increase or prevent work-related stigmatization in association with COVID-19 (C).

We followed the procedures outlined in our study protocol registered a priori on Open Science Framework (https://osf.io/82zav accessed on 3 June 2021).

According to the population–exposure–outcome (PEO) criteria, we designed our systematic search strategy to include studies of the working population (P). While the search strategy was designed to include all studies on new infectious diseases (including SARS-CoV-2, SARS, MERS, influenza virus H1N1, and influenza virus H7N9), this paper focuses on the results of the search on COVID-19 (E). We included all studies on stigmatization of any type (including bullying) in association with work due to COVID-19 assessed with validated or non-validated instruments (O). Furthermore, we included any measures to prevent or deal with occupational stigmatization. Inclusion and exclusion criteria are listed in Table 1.

Additionally, we studied health consequences (O) from the stigmatization of employees (P) due to COVID-19 (E). The PEO for this is shown in Table 2.

We included primary studies (cohort, case–control, case–cohort, RCT, cross-sectional studies, ecological studies, qualitative studies), systematic reviews, dissertations with primary data, or a systematic literature search. Abstracts only, clinical observation studies, books, book chapters, book reviews, comments, corrections, editorials, introductions, forewords, letters, replies, popular science media, and narrative reviews were excluded from this systematic review. Publications with no abstract were included if the title seemed to be relevant to the topic of this review.

We used no geographic or language restrictions, but titles and abstracts not written in English or German were excluded.

### 2.2. Definition of Stigma Forms

Stigmatization forms were differentiated according to the definition by Pescosolido and Martin [5]. Descriptions and examples of each stigmatization form from the included studies are given in Table 3.

### 2.3. Information Sources and Search

We searched the electronic databases Medline (via Pubmed), Embase (via Ovid) and PsycInfo (via EBSCOhost) until 23 October 2020. Moreover, we also searched the search engine of the ZB MED Information Centre for Life Science on 11 December 2020 and included results of a systematic search in the database CINAHL from a cooperating project on stigmatization in physicians and nurses (search date: 1 December 2020). The search strings are shown in the Appendix A. In addition, the grey literature found while searching reference lists were included in this review (checked December 2020). We also used the “citation tracking function” by Google Scholar of the included studies to identify additional studies (January–February 2021).

### 2.4. Study Selection and Data Collection

Search results were imported into an Endnote reference management system database. Study selection was carried out in two steps. First, the screening of titles and abstracts was piloted with a random sample of 500 publications. Four raters (J.L., T.H., A.F. and M.S.) screened the records for eligibility criteria. Interrater reliability was moderate (Fleiss Kappa = 0.42) according to Landis and Koch [42] for the piloting phase. Thereafter, raters were assigned to 2 × 2 rater groups (J.L./A.F. and T.H./M.S.), and the remaining publications were divided equally between the two groups. The titles and abstracts were screened independently by two raters for inclusion and exclusion criteria. Disagreements regarding the inclusion were discussed and studies were included in the full-text screening to err on the side of caution if discussion was not resolved. Interrater reliability was moderate for both rater teams (Cohen’s kappa = 0.49, and 0.51). Second, the screening of all included full texts was carried out by the rater teams. Disagreements were solved by discussion in the research group. If a study was not included based on the full text screening, the reasons for exclusion were reported. The screening of full texts was piloted. For this, 20 randomly chosen publications were distributed to all raters, and interrater reliability was very good (Fleiss Kappa = 0.69). Full texts were divided equally between the two groups and checked for eligibility. Interrater variability was very good for both rater teams (Cohen’s kappa = 0.79, and 0.81).

For the included studies, we extracted the following information from the publications in tables: title, author(s), publication year, study design, characteristics of the study population with job description, time of assessments, exposure, outcome measures and results. Further details regarding the study, such as the adjustment for confounders, ethical clearance, conflicts of interest, and funding sources were also extracted in the comments section of our extraction form. Data extraction was piloted. Data extraction was carried out by one reviewer of the team and checked by the other reviewer of the team (J.L./A.F. and T.H./M.S.).

### 2.5. Rating of Methodological Study Quality (Risk of Bias)

The risk of bias was assessed only for studies reporting stigmatization prevalence (including information on frequency/occurrence of stigmatization) or the prevalence of health consequences. The methodological study quality was determined according to Ijaz et al. [43] and Kuijer et al. [44] with modifications. This method has been used previously in other studies [45].

For the evaluation of the study quality, the following domains were used: (1) recruitment procedure and follow up (in cohort studies), (2) exposure definition and measurement, (3) outcome source and validation, (4) confounding and effect modification, (5) analysis method (methods to reduce research specific bias), (6) chronology, (7) blinding of assessors, (8) funding, and (9) conflict of interest. The quality assessment includes six major domains (1–6) and three minor domains (7–9). Per definition, an overall low risk of bias for the study was assumed if all major domains were rated low. Disagreements were solved by discussion. The methodological quality was independently assessed by at least two reviewers (J.L., A.F., T.H., or M.S.) for each study. Disagreements in quality ratings between the two raters were solved by discussion.

### 2.6. Synthesis of Results and Meta-Analysis

The study results were summarized descriptively and in meta-analyses. For estimating the prevalence of stigmatization, we only considered studies with an adequate sampling design. Thus, we excluded studies characterized by convenience sampling from results synthesis for the reason that assumptions of those studies are not generalizable.

Meta-analyses were performed to estimate the pooled risk of stigmatization on depression and anxiety. The meta-analysis was carried out if at least two primary studies which were comparable in terms of exposure and outcome were present. Due to the heterogeneity of the studies, random effects models were used as the analysis method. Since adjusted and unadjusted risk estimates were close to each other, we decided to include adjusted and unadjusted values in the meta-analysis. The software Stata Version 14.0 (StataCorp, College Station, TX, USA) was used.

## 3. Results

### 3.1. Study Selection

We identified a total of 7452 records from the electronic database and preprint servers of which 167 full texts were assessed for eligibility. Furthermore, 33 additional records were identified from the grey literature search. We included a total of 46 articles in our systematic review. A total of 154 full-text articles were excluded from further consideration. The most common reason for exclusion was that the articles were editorials, comments, opinions, letters, or narrative reviews (*n* = 68). Furthermore, 34 articles were excluded because the exposure was related to other emerging respiratory virus diseases (such as MERS, SARS) and did not include COVID-19. In 30 cases, the outcome did not relate to stigmatization, and in another 10 cases, the study did not include the working population. Further articles were excluded for not being related to the topic (*n* = 5), exposure was not work (*n* = 2), poster presentation (*n* = 1) and being a duplicate (*n* = 1). In addition, despite extensive efforts by our librarian, we were unable to locate the full texts of three records. The literature search and all the reasons for the exclusion of full-text articles are summarized in the PRISMA flow diagram (Figure 1). Moreover, references of all included and excluded studies with the reason for exclusion are listed in the Appendix A.

### 3.2. Study Characteristics

A total of 46 articles examining associations of work-related COVID-19 exposure with the risk of stigmatization were included: 23 articles from 22 cross-sectional studies, 2 longitudinal studies and 11 interview studies. Further, we included six systematic reviews, three study protocols and one intervention study in our qualitative synthesis.

#### 3.2.1. Systematic Reviews

We included six systematic reviews with original studies on work-related stigmatization due to COVID-19 exposure in our literature search [28,30,33,46,47,48]. Four of the reviews [28,30,33,46] only included one original study in their review, which we all identified by our systematic search (original studies: Blake et al. [49], Chatterjee et al. [50], Juan et al. [51], and Mohindra et al. [38]). Joo et al. [47] included the qualitative studies by Kackin et al. [52] and Kalateh-Sadati et al. [53] in their systematic review. Three studies on work-related COVID-19 stigmatization were included by Rahman et al. [48], of which two were excluded by us because they did not meet our inclusion criteria [54,55]. The qualitative study by Fawaz et al. [56] was identified by our search. Information on the original studies is provided in the following tables as well as in the Appendix A.

#### 3.2.2. Quantitative Studies

We included 22 cross-sectional studies (23 articles) [35,36,37,38,40,41,50,51,57,58,59,60,61,62,63,64,65,66,67,68,69,70,71] and two longitudinal studies [39,72] in our systematic review. Most studies were conducted in India (*n* = 5) [38,50,67,68,69] and China (*n* = 4) [41,51,66,72], followed by Vietnam (*n* = 2) [36,40] and Libya (*n* = 2) [58,59]. One study each was conducted in Canada [35], Colombia [63], Egypt [70], Italy [64], Iran [71], Nepal [61,62], Pakistan [37], Singapore [39], the UK [60], and the USA [65]. One study was conducted around the world [57]. The association of COVID-19 with stigmatization was predominately studied in HCWs, medical jobs (such as lab technicians, medical officers), and other jobs related to hospitals (attendants/cleaners, managers/clerks) (19 studies). One study [36] considered HCWs, professional educators and white collar workers, and another study [72] differentiated between governmental/public institutions and private enterprises. Tan et al. [66] investigated returning to work in workers/technical staff and executives, sales and marketing management, and others. Furthermore, Taylor and colleagues [35] investigated the attitudes from the general population towards HCWs, and only included non-HCWs. Another study [57] did not report any information on the work of participants. Besides the study by Said et al. which included only women [70], all studies included females and males. All studies were conducted during the COVID-19 pandemic, of which 13 studies did not give more detailed information concerning the epidemic/pandemic phase. Four studies were carried out during lockdown [36,38,61,62,64]. One study each was performed during the “rapid increase in COVID-19 cases and death”, in the “post-peak phase” [60], at the “highest point” [51], during the “initial phase of containment” [63], or “two weeks after government suspended all public transports” [41]. In addition, one study was carried out after returning to work from lockdown and quarantine during COVID-19 peak [66]. Another study was conducted after the quarantine of employees working in a hospital that was locked down due to a COVID-19 outbreak [40]. Stigmatization was assessed with a wide range of different validated and non-validated instruments. Ten studies measured stigmatization with a single item [41,50,51,57,58,59,60,61,62,66,72]. Two studies each used 2-item [64,70], 4-item [36,37], 12-item [39,40], and 13-item questionnaires [67,68]. One study each used 8-item [35], 19-item [38] and 22-item questionnaires [71]. Yadav and colleagues [69] used an adapted stigma assessment and reduction of impact (SARI) stigma scale. Monterrossa-Castro et al. [63] and Sharma et al. [65] did not provide detailed information on the instrument used.

The majority of studies investigated perceived self-stigma (17 of 24 studies) which was commonly assessed via feelings of discrimination (*n* = 13) [40,51,61,62,63,64,66,67,68,69,70,71,72] and social exclusion (*n* = 5) [36,41,50,67,68]. In addition, four studies investigated perceived aggressive/behaviour/bullying [38,57,61,62,69], and one study used stereotypes [38]. Further, anticipated self-stigma was determined in seven studies [36,37,39,40,64,67,68]. Internalized self-stigma and associative stigma (via social exclusion and aggression/bullying) was investigated in four studies each (associative stigma: [36,57,69,73], internalized self-stigma: [38,39,40,70]). One study investigated public stigma towards HCWs in the general population, measuring stereotypes, discrimination and social exclusion [35].

The majority of studies comprised of convenience samples according to the sampling technique, or because the number of invited persons was not reported (*n* = 17). Only seven studies used an adequate sampling design [39,41,51,58,66,69,71]. All studies were considered of low methodological quality (i.e., high risk of bias). A short description of all studies is given in Table 4. The full extraction table is shown in the Appendix A.

#### 3.2.3. Qualitative Studies

We included 11 qualitative studies and one cross-sectional study with a qualitative part in this systematic review. Studies were conducted in Iran [53,76] and Pakistan [77,78] (*n* = 2 each) as well as in Canada [79], Germany [80], Lebanon [56], Nepal [81], South Korea [82], and Turkey [52] (*n* = 1 each). The USA, Kenya, Ireland and Canada [83] were presented by one study each. Most studies included working males and females (*n* = 7). Two studies only interviewed female nurses [79,82]. Another two studies did not provide any information concerning the gender of participants [53,56]. Nearly all studies were performed in HCWs. Six studies included nurses only, while one study interviewed physicians and nurses [56]. Another study also included the hospital management involved in COVID-19 management in addition to physicians and nurses [78]. Crowe et al. [79] interviewed nurses and employees from high acuity units in an academic teaching hospital. Zolnikov et al. [83] interviewed first responders including HCWs, firefighters, paramedics, police officers, and technicians. Bhatt and colleagues [81] included individuals working at the forefront in the community (HCWs, police officers, school teachers). One study did not give a job description [77]. In addition, we included results of the cross-sectional study by Dye et al. [57] which consists of a qualitative part. For a general description of this study, see Table 5.

Perceived self-stigma was assessed in 11 of 12 studies. In addition, five studies determined associative stigma [52,57,80,81,82]. One study each investigated anticipated self-stigma [78] and internalized self-stigma (use of negative words “feeling dirty”, “feeling “contaminated”) [83].

Full data extraction of studies is provided in Appendix A.

#### 3.2.4. Intervention

We included one study that rapidly developed a free digital learning package for promoting mental health in UK healthcare employees in response to the COVID-19 pandemic. [49]. For a description of the results, see Section 3.4.

#### 3.2.5. Study Protocols

Three study protocols were included in this systematic review. One study protocol outlines the procedure of a scoping review for a systematic literature search on people with disabilities or other vulnerabilities during the COVID-19 pandemic [84]. The review has not been published so far (checked: 22 April 2021).

The second study protocol describes the procedure for a repeated cross-sectional study on the mental health of HCWs and the general public during COVID-19 in Thailand (study name: HOME-COVID-19) [85]. The study aims to assess public stigma towards COVID-19 infections. The results of a single wave have been published in two studies [86,87], but work-related stigmatization due to COVID-19 exposure has not been investigated yet.

The third publication presents a protocol for guideline for “Psychological First Aid (PFA)” in HCWs in Malaysia [88].

### 3.3. Prevalance of Stigmatization with Regard to Occupational Group

#### 3.3.1. Quantitative Studies

For evaluating the prevalence of stigmatization, only studies with an adequate sampling design were considered (*n* = 7). Studies characterized by an adequate sampling design were mostly from China (*n* = 3) [41,51,66]; one study each was from India [69], Iran [71], Libya [58], and Singapore [39]. The responses ranged from 37–92%. All but the study by Tan et al. [66] investigated stigmatization in HCWs, or hospital-related jobs.

Six studies measured perceived self-stigma: Zhu et al. [41] found that 19.5% of the participants felt avoided by family and friends (social exclusion). Yadav et al. [69] described a similar prevalence of 19.3% for perceived self-stigma, of which the majority (70%) experienced rude behaviour by community, followed by racial/obscene/derogatory remarks (32%) and harassment by their landlord and neighbours (32%). In addition, 13% felt harassed by security personnel. Moreover, more resident doctors and nurses perceived stigma than faculty/medical officers. Elhadi and colleagues [58] found that 31.8% of HCWs working in Libyan hospitals felt stigmatized, and this was higher in female than in male HCWs (36.1% versus 28.2%). The study by Tan et al. [66] investigated perceived self-stigma in workers and technical staff, executives, sales and marketing management returning to work after lockdown and quarantine. Perceived discrimination (moderate to very serious) was lower in management and executive staff than in workers/technical staff (0.8% versus 3.5%). Zandifar et al. [71] used a modified HIV Stigma Scale and presented their results on perceived self-stigma (discrimination) using the median without the classification of values. Stigmatization was highest for physicians (median = 29), followed by nurses (26), and technicians (22). Juan et al. [51] investigated psychological distress in hospital staff in association with perceived discrimination during the highest point of the epidemic in China but did not give information on the prevalence. A longitudinal study by Chew et al. [39] measured anticipated and internalized self-stigma using a 12-item healthcare worker stigma scale. The results were presented as means and the authors showed that anticipated and internalized self-stigma was higher at baseline during the COVID-19 pandemic compared to the follow-up after 3 months. Stigma (HWSS total score) was not significantly associated with being exposed to patients with respiratory disease (B = 0.451, *p* = 0.656). Associative stigma was only studied by Yadav et al. [69] and about 11.8% of family members were affected by this. Detailed results of all studies are shown in the extraction tables in the Appendix A.

#### 3.3.2. Qualitative Studies

The results indicate that HCWs and their families are especially prone to stigmatization in daily life. Participants and their relatives reported being harassed, attacked and bullied by neighbours and community who perceived them and their family as “corona-infected”. Moreover, HCWs were not allowed to enter a supermarket [57], or to use public transport/taxi [53,82]. Furthermore, they were asked to leave the apartment [57,81] and were also denied proper food and lodging in hotels [81]. Comments in the news questioned why HCWs go out to eat or to the gym [82]. One female nurse reported that people were yelling at her for being too close and for being outside walking the dog [79]. Another female reported to be sneezed on at very close distance on purpose for enforcing physical distancing at work [57]. Zolnikov and colleagues [83] also showed other aggressive public reactions, including a person pulling down her mask and coughing at the first responders. In two studies, nurses reported that their husbands were not allowed at their work [77,80]. At work, a nurse became the target of criticism (flooding of social network posts) when she was diagnosed with COVID-19 [82]. Moreover, one physician reported being treated as suspicious by other physicians despite negative test results and the strict appliance of personal protective equipment measures [77]. Another participant reported that the work contract of his son who got COVID-19 on duty as a security officer in a bank was not resumed [77].

HCWs/first responders do not only face stigmatization from the community/workplace but also from friends and family [52,53,56,76,79,83]. Most frequently, HCWs reported being “treated like a virus”, and were excluded by family and friends, leading to alienation. Reactions were more exaggerated when it was known that the person worked in the COVID-19 unit. Additionally, in one study, the participant was asked by a family member whether he was short in money caring for COVID-19-patients and was recommended to leave the job [76].

As a result, HCWs hide information about working with COVID-19 patients from family and community [78,82]. Due to the negative impact on social relationships, participants were feeling isolated and annoyed. Some participants also reported increased feelings of frustration and anger [56,82], thinking about quitting [82], or to not work in the COVID-19 ward any further [76]. A detailed extraction of studies is provided in the Supplementary File S5.

### 3.4. Stigmatization from Work-Related COVID-19 Exposure and Health

#### 3.4.1. Descriptive Summary

Associations of work-related stigmatization from COVID-19 exposure with health was investigated in 14 studies [35,39,40,41,50,51,58,59,60,61,63,64,65,68,71,83]. All but one study [83] were quantitative studies. Studies assessed a variety of psychological disorders (Appendix A). Greene et al. [60], Juan et al. [51], Monterrossa-Castro et al. [63], and Zhu et al. [41] used the 7-item Generalized Anxiety Disorders Questionnaire (GAD-7) for assessing anxiety disorders. The authors showed that anxiety was significantly increased in association with being stigmatized. This was also true for depressive symptoms measured with the Patient Health Questionnaire 9 (PHQ-9, *n* = 3) [41,51,60]. The Depression, Anxiety, and Stress Scale (DASS-21) was used in two studies [40,50]. The Hospital Anxiety and Depression Scale (HADS) was also used in two studies [59,61]. Studies found a significant association between psychological symptoms and being stigmatized. In addition, Chew et al. [39], Juan et al. [51], and Zhu et al. [41] studied the experience of traumatic events using the Impact of Event Scale—Revised (IES-R) and found increased traumatic stress in association with being stigmatized due to work-related COVID-19 exposure. Further, perceived self-stigma was associated with a higher risk of post-traumatic stress disorders (PTSD), measured with the PTSD subscale of the International Trauma Questionnaire (ITQ) [60], or the Posttraumatic Stress Disorder 8-item Questionnaire (PTSD-8) [71]. Elhadi and colleagues [58] used an abridged version of the Maslach Burnout Inventory (MBI), and showed that emotional exhaustion was significantly correlated with feeling stigmatized. Higher perceived stress using the Perceived Stress Scale (PSS-10) was shown by two further studies [39,68]. Taylor et al. [35] studied the COVID Stress Syndrome (CSS) with a self-developed 5-item scale. The Quality of Life Scale (ProQOL) was used by Ramaci et al. [64]. Here, perceived and anticipated stigma was related to higher levels of compassion fatigue and burnout as well as lower compassion satisfaction. In addition, Juan and colleagues [51] investigated associations of stigmatization and rejection in the neighbourhood because of hospital workers with obsessive compulsive symptoms using the Yale-Brown Obsessive-Compulsive Scale (Y-BOCS) and the Patient Health Questionnaire 15 (PHQ-15) but found no significant relationship. Further, a significant association between stigmatization and insomnia (measured with the Insomnia Severity Index (ISI)) was found by Khanal et al. [61]. Sharma et al. [65] found increased emotional distress/burnout in association with stigma from community in HCWs but did not describe the instrument used.

In the interviews by Zolnikov et al. [83], participants reported negative feelings, stress, alcohol usage in association with stigmatization. Instruments and results on health effects reported in the original studies are shown in Appendix A.

#### 3.4.2. Synthesis of Results

Four studies reported risk estimates for depression (Figure 2A) and five studies for anxiety (Figure 2B) with regard to work-related stigmatization. The prevalence of depression varied: 13.5% [41,61] over 29.6% [51] to 46.9% [60]. The prevalence of anxiety varied from 18.3% [61], 24.1% [41], 31.6% [51], 39.3% [63] to 47.3% [60].

The results of the meta-analyses indicate a significant association between the work-related stigmatization of HCWs and symptoms of depression and anxiety. In detail, the odds for depression were significantly increased by 74% (OR = 1.74; 95% CI 1.29–2.36, Figure 2A) and anxiety by 75% (OR = 1.75; 95% CI 1.29–2.37, Figure 2B) for HCWs experiencing work-related stigmatization.

### 3.5. Measures to Prevent Work-Related Stigmatization from COVID-19

There was no quantitative study focussing on measures increasing or preventing work-related stigmatization. In the qualitative study by Zolnikov and colleagues [83], participants were interviewed regarding measures which either increase or prevent work-related stigmatization. Good communication among colleagues was considered an important measure to prevent/deal with stigmatization at work. Further, keeping the connection between HCWs/first responders with those outside their professional roles was considered very important. Additionally, education and the dissemination of science-based information related to COVID-19 was deemed an important potential solution to fight stigmatization due to COVID-19 exposure. On the contrary, the distribution of misinformation (e.g., via social media) contributes to higher work-related stigmatization.

Furthermore, the study by Blake et al. [49] introduced a free online tool for sustaining mental health during the pandemic for UK HCWs. The development included a three-step process: 1. “public involvement activities”, 2. “iterative peer review”, and 3. “delivery” (measured as the number of users within 7 days after launch). The package was designed to offer actions team leaders can take for promoting mental health in staff and to provide guidance. The package includes the following topics: psychological impact, psychologically supportive teams, communication, social support, self-care, managing emotions, and further resources. In the section “communication”, social stigma is addressed amongst other things. After the first 7 days of release, the package was accessed 17,633 times. A total of 55 participants completed the evaluation (49 HCWs and six students). The results indicate that the package has a high user satisfaction and that HCWs adopted the guidance in their daily life, e.g., taking further actions to emotionally support colleagues, consideration of psychological first aid training, or calling a telephone helpline. The package is provided at: https://www.nottingham.ac.uk/toolkits/play_22794#resume=1 (accessed on 3 June 2021).

Sulaiman and colleagues [88] developed a protocol for a “Psychological First Aid (PFA)” guideline for Malaysian HCWs affected by COVID-19 or suspected to be infected. The aid is based on the guidelines from the International Federation of Red Crescent Societies (“Look, Listen, Link”) and employs a mobile application and phone calls. The aid is applied in a “Specific, Measurable, Attainable, Relevant, and Realistic Timeframe” (SMART) and contains five stages: (1) input—expert team set up and protocol development, (2) process—PFA-training and service promotion, (3) output—implementation, (4) outcomes—improvement of HCWs health, and (5) aim—evaluation of the PDA protocol. One objective is to encourage HCWs to get support from colleagues and employers, and to minimize stigma (core action: connection with social support). Information on the psychological effects resulting from work-related COVID-19 exposure and assistance for help are provided on the official website and via the promotion of social media, and posters.

## 4. Discussion

In summary, 46 articles examining stigmatization (mostly measured as perceived self-stigma) in the context of work-related COVID-19 exposure (mostly in HCWs) were included in this systematic review. Generally, all included studies indicate that stigmatization occurs as a result of work-related COVID-19 exposure. From a qualitative perspective, HCWs and their families suffer from stigmatization by the community and their work environment. Moreover, several reports indicate that HCWs are also discriminated against and socially excluded by family and friends. As a result, HCWs may not disclose information such as working with COVID-19 patients from their environment to reduce the impact of stigma [78,82]. This may lower feelings of belonging and social integration [89], as well as the usage of professional help [16]. However, selective disclosure may also limit stigmatization and ensure social support [90].

Most quantitative studies were characterized by convenience sampling (17 of 24). Since the results of these studies might be severely biased and do not allow for general conclusions, we did not include them in our result synthesis. Studies with an adequate sampling design (*n* = 7) were considered of low methodological quality, and only reported stigmatization prevalence percentage. Thus, results are hardly comparable and due to missing comparison groups, risk estimates cannot be determined. Descriptively, about 19% of HCWs felt avoided by family and friends [41]. Perceived self-stigma from the community/neighbourhood was perceived by 19% [69], respectively, by 32% [58]. Associative stigma was perceived by 12% [69]. Moreover, in the study by Elhadi et al. [58], female HCWs felt more stigmatized than male HCWs (36.1% versus 28.2%). However, so far, there is no conclusive evidence for gender differences in the perception of stigma [91,92,93]. Furthermore, studies indicate that stigmatization was higher in doctors and nurses as compared to technicians/faculty officers [69,71]. Chew and colleagues [39] showed that stigma was not related to being exposed to patients with respiratory disease. Thus, work-related stigmatization from the community may be directed against an occupational group per se regardless of having actual contact to COVID-19 patients or not (as this is usually not known by the community). Occupational stigma has typically focused on “dirty work” [94]. Thus, working in a hospital per se may be valued as more “dirty” in response to the COVID-19 pandemic than before. Shifts of occupational stigma during the pandemic have been especially observed in service workers. Whereas service work has been conceived as “dirty” before the pandemic, they have been put in the status of a “working hero” nowadays [95].

Associations of work-related stigmatization due to COVID-19 exposure with health were investigated in 14 studies, mainly by measuring symptoms of depression, anxiety, perceived stress, burnout, PTSD, insomnia, obsessive–compulsive behaviour, and somatization. The results for depression (measured with the PHQ-9) and generalized anxiety disorders (measured with the GAD-7) were summarized by meta-analysis. Work-related stigmatization from COVID-19 exposure significantly increases the risk for depression (OR = 1.74; 95% CI 1.29–2.36) and anxiety (OR = 1.75; 95% CI 1.29–2.37). The results are in line with recent systematic reviews on the psychological effects of providing healthcare during viral epidemic outbreaks [25,26]. Here, stigmatization has been shown to be a risk factor for acute and post-traumatic stress and psychological distress in HCWs caring for MERS-CoV2/SARS-patients. Additionally, a recently published cross-sectional study by Hennein et al. [96] on work-related stigmatization experiences in US HCWs shows similar results for major depression (OR = 1.49, 95% CI 1.28–1.74) and generalized anxiety disorders (OR = 1.39, 95% CI 1.19–1.61), although disease prevalence was higher in our pooled analysis. Thus, in addition to the direct consequences of the COVID-19 pandemic, stigmatization further leads to an additional burden for workers.

We identified one qualitative study on conditions increasing/preventing stigmatization at work. The results indicate that communication, education and science-based information present useful tools to prevent work-related stigmatization. In addition, in the UK, a free online tool offers actions that team leaders can take for promoting mental health in staff and to provide guidance [49]. Since stigmatization is shown from diverse sources including the community, workplace and the social environment, anti-stigma strategies should be implemented at different levels including intra- and interpersonal, organizational/institutional, community and governmental/structural levels [97]. In addition, the International Labour Organization [98] developed a guideline to manage work-related psychological burden arising from the COVID-19 pandemic. This includes a section for how to deal with violence and harassment at work (which might be triggered by stigma). The actions are: (1) developing a workplace policy on violence and harassment, (2) implementing measures to protect workers from third party violence, (3) providing clear instruction how to defuse hostile situations, (4) establishing procedures for preventing the discrimination and harassment of workers in general and in particular workers infected with COVID-19, and (5) awareness of domestic violence. Employers should implement these actions for protecting the mental and physical health of the workforce. Furthermore, since research on anti-stigma strategies and their effectiveness is still very limited, we encourage future research to evaluate interventions for preventing work-related stigmatization.

The strengths of this systematic review are the systematic search in three databases with the forward and backward grey literature search of included studies focussing on COVID-19, and the presentation of data extraction in detail. Additionally, we have used a theory-based categorization of different stigma forms providing the basis for tailor-made work-related prevention programs. In addition, we have evaluated the included studies for their methodological quality for detecting a potential bias of the results. Study quality was a major problem, and we encourage future research to incorporate higher quality standards for sampling of the study population, the use of validated questionnaires, and the use of adequate comparison groups. The use of cohort studies as a study design is desirable. Moreover, since studies were predominately limited to HCWs, we further encourage future research on diverse occupations.

A limitation of this systematic review is the inclusion of publications from all over the world without taking cultural differences into account. Stigma is considered a cultural phenomenon, and global differences in stigmatization have been described [99]. Thus, we encourage future research to include cultural differences in the study of stigmatization due to work-related COVID-19 exposure and associated health consequences. In addition, since we searched electronic databases until October 2020, we were not able to investigate the influence of COVID-19 self-testing and the availability of vaccination on stigmatizing attitudes or on perceived stigma. Future studies on this topic are desirable.

## 5. Conclusions

As far as we are concerned, this is the first systematic review with a comprehensive overview of studies on work-related stigmatization in association with COVID-19 only. However, the scientific value of most studies is very limited, because the study population was recruited by convenience sampling, or because studies were characterized by low methodological quality. Additionally, studies used different techniques for assessing stigmatization, and a comparison group was missing. Thus, we encourage future research for adopting higher methodological standards for conducting and reporting of studies. This would provide a good basis for a review update limited to adequate studies in the future.

Regardless of the difficulties of reliably quantifying the extent of stigma due to methodological deficiencies in the studies, the stigmatization of occupations with contact to COVID-19 or suspected patients is a relevant problem. We found clear evidence of the psychological consequences of COVID-19-related stigmatization for depression and anxiety disorders. For promoting workers’ health, anti-stigma strategies and psychological support should be implemented in the workplace.

## Figures and Tables

**Figure 1 ijerph-18-06183-f001:**
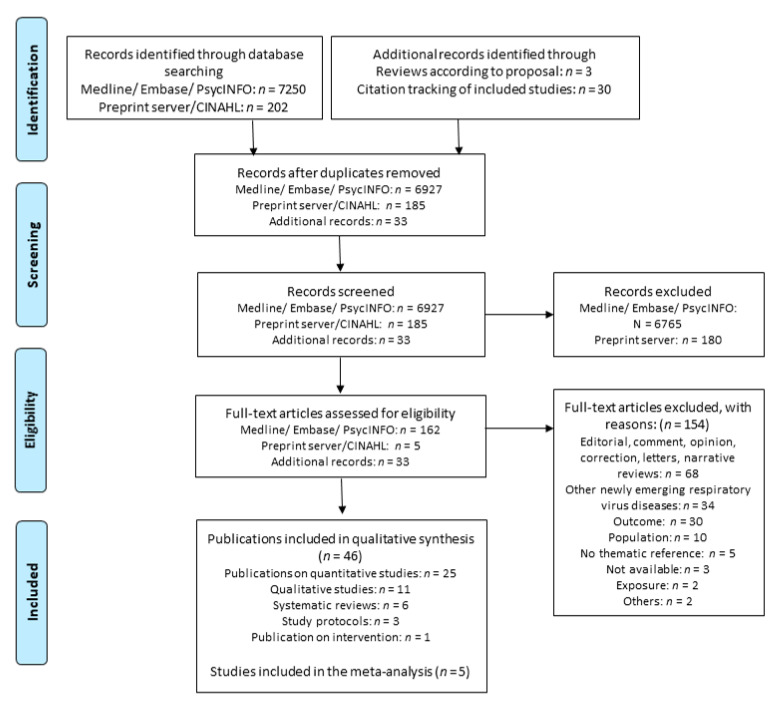
PRISMA flow diagram. Adapted from: Moher D, Liberati A, Tetzlaff J, Altman DG, the PRISMA Group (2009). Preferred reporting items for systematic reviews and meta-analyses: the PRISMA statement. PLoS Med 6(6): e1000097, doi:https://doi.org/10.1371/journal.pmed1000097.

**Figure 2 ijerph-18-06183-f002:**
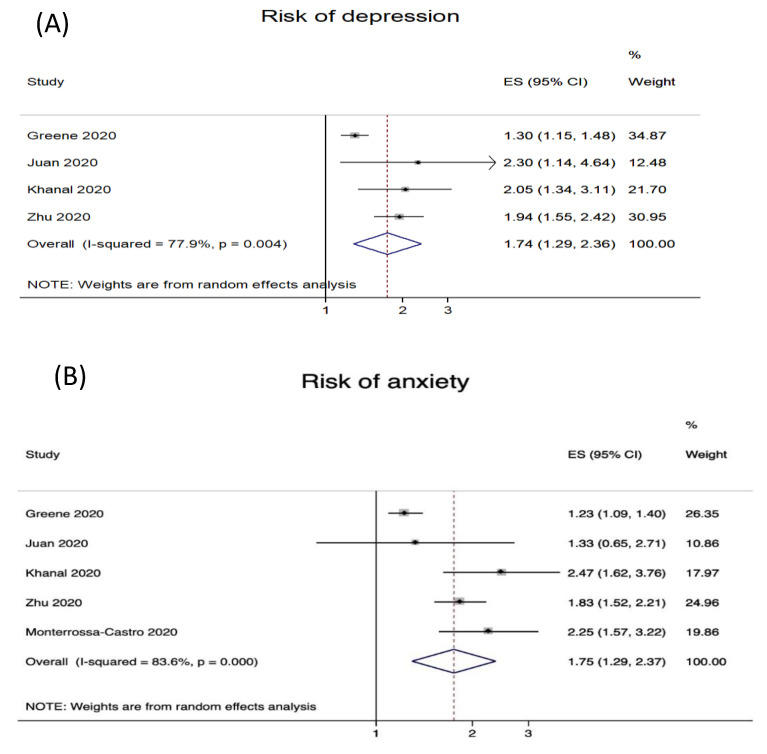
Risk of depression (**A**) and anxiety (**B**) in association with work-related stigmatization from COVID-19 exposure.

**Table 1 ijerph-18-06183-t001:** Inclusion and exclusion criteria for the research question concerning work-related stigmatization as outcome (research question A and C).

Category	Inclusion	Exclusion
Population	General working population (all sexes)	Children and youth, unemployed persons or persons in non-paid employment, pensioners, and persons over 70 years
Exposure	SARS-CoV-2 (A)	Other infectious diseases such as HIV and tuberculosis ^1^
Outcomes	All stigmatization forms (including bullying) in association with work (A); measures that prevent work-related stigmatization (C)	-

^1^ Results on new respiratory virus diseases other than COVID-19, i.e., SARS, MERS, influenza virus H1N1, and influenza virus H7N9 were excluded in the further study process.

**Table 2 ijerph-18-06183-t002:** Inclusion and exclusion criteria for the research question concerning health consequences of stigmatization (research question B).

Category	Inclusion	Exclusion
Population	General working population (all sexes)	Children and youth, unemployed persons or persons in non-paid employment, pensioners, and persons above 70 years
Exposure	Work-related stigmatization (all forms) due to COVID-19	-
Outcomes	Physical and mental health, stigma-reducing strategies	-

**Table 3 ijerph-18-06183-t003:** Definition of stigma forms of included studies.

Stigma Forms	Description	Examples from Included Studies
Public stigma	Endorsement of stereotypes, prejudices and discrimination against a group, that holds a specific characteristic. In surveys, the public is asked.	Stereotypes: e.g., “Healthcare workers who work in hospitals are likely to have COVID-19” [35]. Discrimination: e.g., “Healthcare workers should have some restrictions on their freedom [35]. Social exclusion as a component of discrimination: e.g., “I do not want to be around someone who works in a healthcare setting [35].
Associative stigma	Endorsement of stereotypes, prejudices and discrimination against a group which is transferred to relatives. In surveys, the public or affected persons are asked.	Discrimination: e.g., “Relatives being alienated because employment related to COVID-19” [36]. Social exclusion as a component of discrimination: e.g., “People would avoid my family members because of my job” [37]. Aggressive behavior/bullying as a component of discrimination: e.g., “Have verbally abused (…) or physically assaulted (…) my family members” [38].
Self-stigma (internalized)	Internalization, i.e., perception and transfer of stereotypes and devaluations to the own person. In surveys, affected persons are asked.	e.g., “Felt guilty about possibly exposing family, community and peers to infection” [38]; “feeling inferior to others due to occupation” [39].
Self-stigma (perceived)	Belief that “most people” will devalue and discriminate the stigmatized. In surveys, affected persons are asked.	Stereotypes: e.g., “People do not trust me and worry that I might infect them” [38]. Discrimination: e.g., „People feel uncomfortable when I am around” [40] Social exclusion as a component of discrimination: e.g., “Family members and friends have avoided contact with me because of my work?” [41]. Aggressive behavior/bullying as a component of discrimination: e.g., “People have verbally abused me or physically assaulted me” [38].
Self-stigma (anticipated)	Expectation of experiencing prejudice and discrimination due to a specific characteristic. In surveys, (potentially) affected persons are asked.	Fear, perceived consequences: “People would avoid me because of my job” [37].

**Table 4 ijerph-18-06183-t004:** Summary of included quantitative studies.

First Author Risk of Bias	Region Study Design Comments	Population	Stigmatization Form Assessment	Time of COVID-19 Pandemic
Chatterjee et al. [50] High risk	India CS Convenience sample	Physicians	Self-stigma (perceived) 1-item	Not specified
Chaudhary et al. [37] High risk	Pakistan CS # invited n. r.	Clinical oral HCWs Non-clinical oral HCWs from 10 different dental hospitals	Self-stigma (anticipated) Associative stigma 4-items	Not specified
Chen et al. [72] High risk	China Longitudinal # invited n. r.	Government/public institution/institutions/state-owned, enterprises, private enterprise staff or individual business	Self-stigma (perceived) 1-item	Baseline: rapid increase in COVID-19 cases and related deaths; follow-up: authorities relaxed lowdown
Chew et al. [39] High risk	Singapore Longitudinal Response: 49.2%	Medical residents in training (medical and surgical)	Self-stigma (anticipated) Self-stigma (internalized) 12 items (Healthcare Workers Stigma scale, HWSS)	Not specified
Dang et al. [36] High risk	Vietnam CS Convenience sample	HCWs, professional educators, white collar workers, students, others	Self-stigma (perceived) Self-stigma (anticipated) Associative stigma 4-items	Data collection one week after social distancing and lockdown was ordered by government
Do Duy et al. [40] High risk	Vietnam CS # invited n. r.	Clinicians, nurses, others	Self-stigma (perceived) Self-stigma (anticipated) Self-stigma (internalized) 12 items (adaption of Berger’s HIV Stigma Scale)	Lockdown of workplace because of COVID-19 outbreak-> all employees required to quarantine for 23 days. Data collection after quarantine
Dye et al. [57] High risk	Worldwide CS Convenience sample	n.r.	Self-stigma (anticipated) Associative stigma 1 item	Not specified
Elhadi et al. [59] High risk	Libya CS Convenience sample	HCWs (doctors and nurses) from 15 hospitals working during the outbreak period	Self-stigma (perceived) 1 item	Not specified (but during civil war)
Elhadi et al. [58] High risk	Libya CS Response: 88.7%	HCWs working in either surgery, internal medicine, intensive care, or emergency departments	Self-stigma (perceived) 1 item	Not specified (but during civil war)
Greene et al. [60] High risk	UK CS Convenience sample	Frontline health and social care workers working in a variety of healthcare roles in UK hospitals, nursing or care homes, and community settings	Self-stigma (perceived) 1 item	During COVID-19 pandemic (post-peak phase of the initial COVID-19 wave in the UK)
Juan et al. [51] High risk	China CS Response: 91.2%	hospital staff from five national COVID-19 designated hospitals (working in isolation ward, general ward)	Self-stigma (perceived) 1 item	Study period corresponds with the highest point of the COVID-19 epidemic in China
Khanal et al. [61,62] High risk	Nepal CS # invited n. r.	Nurses, doctors, paramedics, laboratory staff, pharmacists, public health professional currently working in COVID-19 management	Self-stigma (perceived) 1 item	During lockdown
Mohindra et al. [38] High risk	India CS Convenience sample	Doctors, nurses, hospital attendants, sanitation attendants, others working at the hospital	Self-stigma (perceived) Self-stigma (internalized) 19 items (adapted from Ebola epidemic questionnaire [74])	During lockdown
Monterossa-Castro et al. [63] High risk	Colombia CS # invited n. r.	General Practitioners	Self-stigma (perceived) (questions not described)	Responses to “the 24–30 March period, when the country was in a health emergency, in the initial phase of containment”
Ramaci et al. [64] High risk	Italy CS Convenience sample	Nurses and doctors	Self-stigma (perceived) Self-stigma (anticipated) Questionnaire adapted from HIV/AIDS/drug users questionnaire [75]	During national lockdown
Said et al. [70] High risk	Egypt CS (controlled) Convenience sample	Nurses from triage hospital and from a hospital with no triage or isolation	Self-stigma (perceived) Self-stigma (internalized) 2 items (from US National Centre for Posttraumatic Stress Disorder 2020 and “MERS-CoV staff questionnaire”)	Not specified
Sharma et al. [65] High risk	USA CS Convenience sample	HCWs caring for COVID-19 patients (intensive care unit): physicians, nurses, respiratory therapists, advanced practice providers	Self-stigma (perceived) Not reported	Not specified
Tan et al. [66] High risk	China CS Response: 50.9%	Members of the workforce who returned to work: workers, and technical staff, executives, sales and marketing, management and others>	Self-stigma (perceived) 1 item	Returning to work after lockdown and quarantine in Chongqing, during the peak of the COVID-19 epidemic when strict infection control was in place
Taylor et al. [35] High risk	Canada, USA CS # invited n. r.	Non-HCWs	Public stigma 8 items	Not specified
Uvais et al. [67,68] High risk	India CS Convenience sample	Physicians working in hospitals	Self-stigma (perceived) Self-stigma (anticipated) 13 items (Perceived Stigma Scale)	Not specified
Yadav et al. [69] High risk	India CS Response: 36.6%	HCWs	Self-stigma (perceived) Associative stigma Adapted Stigma assessment and reduction of impact (SARI) Stigma scale	Not specified
Zandifar et al. [71] High risk	Iran CS Response: 92%	HCWs engaged in the field of diagnostic and treatment of COVID-19 patients working in 9 general hospitals (physicians, nurses, technicians)	Self-stigma (perceived) 22-items (adopted from the HIV Stigma Scale)	Not specified
Zhu et al. [41] High risk	China CS Response: 77.1%	HCWs from hospital directly providing services to confirmed or suspected COVID-19 patients (physicians, nurses, technicians)	Self-stigma (perceived) 1 item	COVID-19 outbreak (2 weeks after the authority in Wuhan suspended all public transport)

# invited n. r. = number of invited participants was not reported, CS = cross-sectional study, longitudinal = longitudinal study design.

**Table 5 ijerph-18-06183-t005:** Summary of included qualitative studies.

Study	Region	Population	Stigmatization Form Assessment	Time of COVID-19 Pandemic
Bhatt et al. [81]	Nepal	Teachers, students, security personnel, head of household, leaders, health workers, homemaker, others	Self-stigma (perceived) Associative stigma Interviews and focus group discussions	Not specified
Crowe et al. [79]	Canada	Critical Care Registered Nurses (CCRN) providing direct patient care in the intensive care and high acuity units in an academic teaching hospital	Self-stigma (perceived) Interviews	During the initial phase of the COVID-19 pandemic
Dye et al. [57]	Worldwide	Not reported	Self-stigma (perceived) Associative stigma Open-ended question	Not specified
Fawaz et al. [56]	Lebanon	Nurses and physicians working at various COVID-19 units	Self-stigma (perceived) Semi-structured interviews	Being quarantined following occupational COVID-19 exposure
Feroz et al. [78]	Pakistan	Key informants KIIs (senior management and hospital leadership, directly or indirectly involved with the management of COVID-19 patients)	Self-stigma (anticipated) Semi-structured interviews and a purposive sampling approach	Not specified
Hien et al. [80]	Germany	Nurses in clinics and retirement homes	Self-stigma (perceived) Associative stigma Interviews	Not specified
Kackin et al. [52]	Turkey	Nurses caring for COVID-19 patients	Self-stigma (perceived) Associative stigma Semi-structured interviews	Not specified
Kalateh-Sadati et al. [53]	Iran	Nurses working in hospitals specified for COVID-19 treatment	Self-stigma (perceived) Interviews	Not specified
Lee et al. [82]	South Korea	COVID-19-designated hospital nurses providing direct care for patients	Self-stigma (perceived) Associative stigma In-depth interviews	Not specified
Reazee et al. [76]	Iran	Nurses working fulltime in COVID-19 wards	Self-stigma (perceived) Associative stigma Interviews	Not specified
Rizvi Jafree et al. [77]	Pakistan	Not reported	Self-stigma (perceived) Semi-structured interviews	COVID-19-affected families admitted at three government-allocated hospitals
Zolnikov et al. [83]	Canada, Ireland, Kenya, USA	First responders/HCWs: nurses, physicians, firefighters, paramedics, police officers, nurse technicians, behavioural therapists, orthodontists, dialysis technicians, technicians in medical surgery, data specialists, emergency medical technicians	Self-stigma (internalized) Semi-structured interviews	Not specified

## Data Availability

We did not collect own data for this systematic review, and our analyses are based on already published data. Data presented in this study are available in the supplementary material (i.e., extraction tables).

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
