# Peer review of "Stigmatization from Work-Related COVID-19 Exposure: A Systematic Review with Meta-Analysis"

_ijerph, 2021, doi:10.3390/ijerph18126183_

Round 1
Reviewer 1 Report
General comments:
I believe that the topic of the manuscript is interesting and hasn´t been studied in the last months
The explained limitations are very important for the design of the study and the interpretation of the results.
Specific comments:
- Writing
The writing, structure and organization of the manuscript is in accordance with the guidelines.
- Title
The title reflects the content and problem studied.
- Abstract
The abstract reflects the manuscript
- Key Words
The keywords are representative of the subject studied and exposed. I would add nursing
- Background
A state of the art is made in relation to the study and describe the rationale for the review in the context of what is already known.
The authors describe the objective
- Methods
The authors provide an explicit statement of the questions addressed with reference to Population-Exposure-Outcome (PEO)
All information sources and the possible biases are described
- Findings
Results show all Prisma points. Result are relevant
- Discussion
Limitations aren´t exposed
- Application to Pratice
The practical application of this investigation isn´t explained.
- References
The references used are current.
Author Response
Dear reviwer,
many thanks for your kind words and your suggestions. We added „nursing“ to the keywords (line: 32). We have included a section on strength and limitation (starting line 564). Furthermore in the discussion, we describe that stigmatization is a problem and we give future directions for research for practical application.
Reviewer 2 Report
Reviewer's comments on the manuscript entitled “Stigmatization from work-related COVID-19 exposure: a systematic review with meta-analysis” (manuscript ID: ijerph-1235285). The aim of this systematic review was to provide a comprehensive overview of COVID-19-related stigmatization across occupational classes. The manuscript itself is interesting and meticulously written. First of all, the comprehensively written research methodology should be truly appreciated. The interestingly developed research results are really exceptional. In addition, a great deal of effort put into preparation of this paper is quite impressive. I would like to congratulate the authors on publishing such a work. I have no major comments as to the content of the manuscript. To my mind, this paper has been developed very well. My only suggestion is that the aim of the work should be written in the past tense (line: 78).
Author Response
Dear reviewer,
many thanks for your kind words and your suggestion. We used the past tense to describe the aims of the study (line 87).
Reviewer 3 Report
This work represents a well-conducted, comprehensive, international literature review on the topic of stigmatization as a result of work-related COVID-19 exposure. In addition, the article comprises a meta-analytic part estimating the risk for depression and anxiety as a consequence of stigmatization. The manuscript is well-written and deals with a relevant topic. The authors extracted 46 relevant articles showing the indirect social consequences of the COVID-19 pandemic in terms of stigmatization while different forms of stigma were considered. As a main result, stigmatization seems to be a large problem for several working groups in the pandemic. Further, an increased risk for anxiety and depression in stigmatized individuals was found. Anti-stigma strategies were suggested. The results of this review may provide new insights and impulses for future prevention approaches. Another positive feature of this work is that the authors provide a lot of supplementary material. Overall, this work is of high quality covering many aspects of stigmatization including the evaluation of methodological quality of studies. However, please find below some minor remarks that could be helpful in improving the manuscript.
Introduction:
- At the end of this section, the aims of the review could be expanded by the meta-analytic part.
- The aspect of bullying could be integrated in the introduction.
Methods:
- Line 165: Please correct “Methodological”.
- Should the term “prevalence of stigmatization” be changed into “frequency/occurrence of stigmatization”? The term “prevalence” might be misleading in the context of stigmatization.
Results/Figures/Tables:
- The solution/quality of Figure 1 may be too low. Maybe, the authors can provide an enhanced version of this figure.
Discussion:
- Please do not repeat/report results without interpretation or embedding them into current state of research.
- What were the strengths and limitations of this review?
- Maybe, the authors could provide more future directions.
Author Response
Dear reviwer,
many thanks for your kind words and your helpful suggestions. Please see attached your reply to your suggestions.
